# Individual and Contextual Factors Associated with Malaria among Children 6–59 Months in Nigeria: A Multilevel Mixed Effect Logistic Model Approach

**DOI:** 10.3390/ijerph182111234

**Published:** 2021-10-26

**Authors:** Phillips Edomwonyi Obasohan, Stephen J. Walters, Richard Jacques, Khaled Khatab

**Affiliations:** 1School of Health and Related Research (ScHARR), University of Sheffield, Sheffield S1 4DA, UK; s.j.walters@sheffield.ac.uk (S.J.W.); r.jacques@sheffield.ac.uk (R.J.); 2Department of Liberal Studies, College of Administrative and Business Studies, Niger State Polytechnic, Bida Campus, Bida 912231, Nigeria; 3Faculty of Health and Wellbeing, Sheffield Hallam University, Sheffield S10 2BP, UK; k.khatab@shu.ac.uk

**Keywords:** malaria, fever, *Plasmodium falciparum*, *Falciparum vivax*, under-five, determinants, risk factors

## Abstract

Background/Purpose: Over the last two decades, malaria has remained a major public health concern worldwide, especially in developing countries leading to high morbidity and mortality among children. Nigeria is the world most burdened malaria endemic nation, contributing more than a quarter of global malaria cases. This study determined the prevalence of malaria among children at 6–59 months in Nigeria, and the effects of individual and contextual factors. Methods: This study utilized data from 2018 Nigeria Demographic and Health Survey (NDHS) involving a weighted sample size of 10,185 children who were tested for malaria using rapid diagnostic test (RDT). Given the hierarchical structure of the data set, such that children at Level-1 were nested in communities at Level-2, and nested in states and Federal Capital Territory (FCT) at Level-3, multilevel mixed effect logistic regression models were used for the analysis. Results: The proportion of children 6–59 months of age in Nigeria that had malaria fever positive as assessed by RDTs was 35.5% (3418/10,185), (CI: 33.9–37.1). Kebbi State had 77.7%, (CI: 70.2–83.5), which was the highest proportion of 6–59 months who were malaria positive, next in line was Katsina State with 55.5%, (CI: 47.7–63.1). The Federal Capital Territory (FCT), Abuja had the proportion of 29.6%, (CI: 21.6–39.0), malaria positive children of 6–59 months of age. Children between the age of 48 and 59 months were 2.68 times more likely to have malaria fever than children of ages 6–11 months (AOR = 2.68, 95% CI: 2.03–3.54). In addition, children from the rural area (AOR = 2.12, 95% CI: 1.75–2.57), were more likely to suffer from malaria infection compared to children from urban area. Conclusion: The study identified some individual and contextual predictors of malaria among children in Nigeria. These factors identified in this study are potential areas that need to be considered for policy designs and implementations toward control and total elimination of malaria-related morbidity and mortality among children in Nigeria.

## 1. Introduction

The past twenty years and up till now, malaria has persisted as a major global public health concern [1,2], with over 300 million cases reported in 2018 [3] and has remained a leading cause of morbidity and mortality. Low and medium-income countries (LMIC), especially in Sub-Sharan Africa (SSA), contribute to more than 80% of the global malaria burden [4,5]. Nigeria, with a population of over 200 million people, is the world most burdened malaria endemic nation, contributing more than a quarter of global malaria cases. The risk of malaria infection cut across all age segments with women, (especially the pregnant), and children (especially under-five years), the most vulnerable. Malaria is a deadly disease that kills an estimated number of 30 children every hour worldwide. There were great commitments by governments and global partners to end malaria induced mortality and morbidity by 2020 [6]. In 1998 Roll Back Malaria (RBM) initiative was established by World Health Organization (WHO) in conjunction with United Nations Children Fund (UNICEF), in partnership with some financial bodies and heads of governments across the United Nations (UN) to reduce malaria induced under-five mortality by 50% in 2010 through prompt diagnosis, treatment, and use of insecticides treated nets [7]. These efforts targeted primarily to improve the health-related quality of life of the child [7]. Furthermore, by 2018 there was a renewed commitment by some commonwealth nations to prevent more than 650,000 deaths arising from malaria infections by 2023 [8].

Between 2001 and 2014, Nigeria has implemented four National Malaria Strategic Plans (NMSPs), with the most recent, which ended in 2020 (2014–2020), was aimed at reducing malaria-related deaths to zero by 2020 [9]. Unfortunately, this was far from being achieved. However, to scale up the intervention strategies through evidence-based data, Nigeria has conducted three nationally representative surveys with the baseline survey conducted in 2010, followed up in 2015, and the third incorporated into 2018 Nigeria Demographic and Health Survey [9,10,11].

The transmission of malaria parasites has its root in the socio-cultural and economic statuses of the people [12]. Studies have showed that age of the child [13,14,15,16], birth order [13,14], breast feeding status [13], anaemia status [5,14], that the child slept under bed net [15,17], maternal education [5,16,18], body weight status [13], age of household head [5], improved source of drinking water [14], place of residence [13,14,15,19], household socioeconomic factors [13,16,20], and regional variations [14,15,21,22] are significant predictors of the risk of malaria infection in under-five years of age in SSA, and in Nigeria.

Effective control of malaria in Nigeria will require strategies that identify areas and characteristics of people that are highly vulnerable to malaria infections leading to the development of plans and the implementation of policies to reach them. This study complements the findings in previous studies to show potential effects of contextual variables at both cluster and state levels [2,11,18]. Therefore, this study is aimed at establishing the prevalence of malaria across the states and federal capital territory, and to examine the individual- and contextual-level predictors of malaria fever among children 6–59 months of age in Nigeria.

## 2. Materials and Methods

### 2.1. Study Area

Nigeria is a country located in West Africa sharing borders with Cameroon, Niger, Benin Republic and the Atlantic Ocean with a total area of 923,768 square kilometres [9,10,11]. The Nigeria population grew from over 140 million people as of 2006 population census [23,24] to more than 180 million people in 2016 and is expected to rise to over 260 million people by the year 2030 with an estimated annual national growth rate of 2.38% making her the most populous black nation in the world [24]. The population density for Nigeria was estimated to about 215 people per square kilometre in 2018 from approximately 194 people per square kilometre in 2015 [25]. The country has variety of ethnic groups of over 250 [23] with different dialects and customs. The three major ethnic groups with a population of over 68% are the Fulani/Hausa, Yoruba, and Igbo, while the Edos, Ijaw, Kanuri, Ibibio, Ebira, Nupe, Tiv and other minority ethnic groups accounted for 32% [23,26]. Nigeria has 37 administrative divisions (36 states and the Federal Capital Territory (FCT) which are classified into six geopolitical zones [15]. The 37 political, administrative areas are sub-divided into 774 local government areas (LGAs), and each of the LGAs are divided into wards with each LGA having between 10 and 15 political wards [27].

### 2.2. Source of Data

This study is a secondary analysis of two independent nationally representative cross-sectional surveys data sets, such that the 2018 National Human Development Report (known as NHDR 2018) [28] data set is incorporated into the 2018 Nigeria Demographic and Health Survey (otherwise known as 2018 NDHS) [11]. The human development index (HDI) and multi-dimensional poverty index (MPI) from the NHDR 2018 were extracted to serve as the state variables.

### 2.3. Sampling Techniques

NDHS, being the primary data set for this study, samples were selected separately from each stratum using a two-stage stratified cluster design on each stratum derived from the 2006 census identified enumeration demarcation. At the first stage, 1400 enumeration areas (EAs) were selected as sampling units. At the second stage, 30 households were randomly selected from each EA using equal probability sampling, leading to a target total sample of 42,000 (30 × 1400) households used for the survey. In total, 11 EAs were excluded from being captured because of in-security. Therefore, a total of 41,668 households was earmarked for sampling, but only 40,427 households representing a response rate of 99.4% were finally captured in the survey [11].

The target groups for the survey were women age 15–49 years from all the randomly picked households and men age 15–59 years in one-third of all randomly selected households across Nigeria [11]. Besides, children 6–59 months in 14,000 households (i.e., in the households where men questionnaire was administered) had their blood sample taken for malaria (via rapid diagnostic test—RDT), anaemia, and genotype testing. In addition, 75% of the children tested for malaria using RDT, were randomly selected for confirmatory test for malaria using macroscopic blood smear in a laboratory. Only 97.2% of all eligible children for RDTs test were successfully captured, resulting to 2.8% missing values. In this study therefore, the unit of analysis was children 6–59 months of age in Nigeria. Further details of sampling procedures have been published elsewhere [11].

### 2.4. Outcome Variable

In 2018 NDHS diagnostic tests for malaria parasite were carried out for children 6–59 months of age in approximately a third of the selected households. Children in this age bracket have not developed enough immunity as such and they are more prone to contracting malaria [29]. Two testing methods were adopted:(i)Rapid diagnostic tests (RDTs) were conducted on blood samples taken from pricking the finger or heal of the child using SD Bioline Ag Pf (HRP-II). The RDTs detect the Histidine-rich protein-II (HRT-II) human whole blood (antigen). The results were either classified as positive or negatives for Pasmodium falciparum (Pf).(ii)Laboratory microscopy investigation on thick blood smears was performed for a three quarter of the households where RDTs was done. Malaria results were also classified as either positive or negative.

There was a strong positive correlation 0.581 (95% CI: 0.57–0.60; *p* < 0.0001), between these two test results. However, the microscopy laboratory test was performed essentially as a confirmatory test for the RDT that was carried out on the field [9,10,11]. Therefore, in this study, the malaria status of children 6–59 months of age in Nigeria using RDTs was classified as “one” if the result was positive and “zero” when the result was negative.

### 2.5. Independent Variables

The predictor variables considered for this study were identified from previous scoping reviews [1,30,31], and categorized in line with previous studies, and as reported in NDHS 2018 final report [21]. Two files in 2018 NDHS, children under-five years (KR), and household members (PR), were merged using a common identifier to obtain variables that meet this analysis [32]. The health status (healthy or not) of a child in a population is a function of interrelated factors at both individual and environmental level [33]. These factors are called determinants and are divided into three main groups: individual characteristics; physical and social; and health services [33]. As for individual determinants, these are on one hand, biological uncontrolled inherent traits from birth that distinguish the health status of one child from another, such as age, sex, parental affiliation. On the other hand, they could be behavioural factors that pertains to the child which are subject to modification through some control measures such as diet, immunization, or food supplements, etc. Other determinants of child’s health outcomes relate to the household’s physical and social environment [33]. In this study, the determinants of the risk of malaria infection among children 6–59 months of age in Nigeria were grouped into child-, parental- household- cluster-, and area-related variables. The definitions and classifications of these variables are given as follows:

#### 2.5.1. Child-Related Variables

Age of the child was classified into quintiles of one year interval each, with the reference group (6–11 months) corresponds to the period the child had not begun to walk, 12–23 months is another important landmark (completion of immunization), by 24–36 months, some children would have had another sibling following, 36 months and above connotes immediate preschool age. Further, sex of the child, maternal perceived birth size of the child classified into, large, average, and small categories. Birth order of the child defines a child’s rank among other children of same mother. However, the effect of this on child’s health outcome is not well established [34]; duration of breastfeeding, breast milk is very important to a child in the first few years of life [35]. It gives the child all the nutrients needed, and thus strengthens the child’s natural immunity against diseases, this was classified as: (i) ever breastfed, not currently breast-feeding (ii) never breastfed (iii) still breastfeeding; whether the child had taken iron, vitamin A, and deworming treatment in the last 6 months before. In addition, also considered were if a child had diarrhoea, fever, or acute respiratory infection (ARI), 2 weeks before survey. Nutritional status was derived through a composite index computed from the four nutrition indicators (stunting, wasting, underweight, and overweight), using the composite index of anthropometric failure (CIAF) [36,37,38]. Children with no trace of anthropometric failure were classified as “well-nourished” and those that have at least one of the four indicators were classified as “poorly nourished”; anaemia status derived from the record of whether the child was mildly, moderately, or severely anaemic and were collapsed into one group, such that a child with haemoglobin level less than 11.0 g per decilitre was regarded as being “anaemic” and classified as “one”, otherwise, for 11.0 g or more was “not anaemic” and classified as “zero”.

#### 2.5.2. Parental-Related Variables

Mother’s age (year) at last birthday was classified as reported in similar studies to distinguish younger mothers (below 25 years), middle aged, and older mothers, above 34 years [39,40,41]. The mother’s age when she had her first child was also important. Younger mothers lack baby-care experience, and their babies are very likely to have poor health outcomes. Age groups of mothers are often supported in 10 s of years brackets [42], age at first birth was classified as (i) 10–19 years, (ii) 20–29 years, and (iii) 30 years+. Mother working status, paternal working status, and mother’s educational status is very important in maternal and child health. Maternal literacy can enhance the level of knowledge, altitude, and practice (KAP) of some common childhood diseases. Paternal education status is worthwhile investigating the role it plays in child’s health outcome. This is likely going to be moderating the effect of maternal characteristics on child’s health. Both maternal and paternal education statuses were classified as: (i) no education, (ii) primary, (iii) secondary and above; marital status (never in union, in union, widow/divorced/separated); mother slept under mosquito net; mother’s body weight status (kg/m2), (i) underweight when BMI <18.5 (kg/m2), (ii) healthy when BMI is 18.5–25.0 (kg/m2), and (iii) overweight/obese when BMI >25.0 (kg/m2). Mother’s anaemia status, and maternal anaemia statuses (whether mildly, moderately, or severely anaemic), were collapsed into one group, such that mild, moderate, or severe were regarded as “anaemic” and classified as “one”, otherwise, it was “not anaemic” and classified as “zero”. Number of antenatal care visit mother attended during the child’s pregnancy was also provided. The WHO as at the time of the survey for this study recommended a minimum of four visits [11,43]. Maternal autonomy was derived using the composite index score from the mother’s level of participation in decision concerning her own health, large purchases, and visits to family [36]. The score ranged from three to six such that three and four represent “less” autonomy, and five and six represent “more” autonomy. Maternal ethnicity affiliation was classified into four groups; and maternal religion status, since it defines people’s belief system and can affect several maternal and child’s health-related outcomes [44].

#### 2.5.3. Household-Related Variables

Household wealth status is the measure of household economic status derived as the composite score from some durable goods using principal component analysis (PCA). The wealth status was originally classified into quintiles. In this study, three categories were derived, such that poor and poorest groups were collapsed into poor, while rich and richest were collapsed into rich categories [41,45]; household had mosquito bed net; household number size measures the number of people that stay in a household, and was originally a scale variable, but for the purpose of this analysis, the variable was classified into four categories in line with average typical family sizes of four to five members in Nigeria [11]; number of bedrooms in household, though in the survey the number of rooms available for sleeping in a household were given in scale values with less than 10% indicated having five or more rooms for sleep. A typical building in Nigeria has either one-, two-, three-, or four-bedrooms, hence in this analysis, we categorized the variables into five groups; number of children under-5 year in household were also treated as categories as in previous studies [2,45]; improved source of drinking water, toilet facilities, floor materials, roofing materials, and wall materials. These materials are either natural, rudimentary, or finished. Unimproved are the natural and rudimentary, and the improved are the finished materials [46]; sex of household head; household head age (years) in groups. In some developing countries, the age status of the household head is very important as it relates to maturity with which final decision making are taken, particularly on the issue of health. This was also classified as, ((i) less than 34 years, (ii) 35–44 years, (iii) 45–55 years (iv) 56 years +); whether the household had electricity. The presence or absence of electricity in household can impact food storage, cooling at certain season can affect the health of children in particular; type of cooking fuel; under-five slept under a bed net last night, (i) no child, (ii) all children, (iii) some children, (iv) no net in the house [11].

#### 2.5.4. Cluster-Related Variables

Proportion of cluster’s household with no bed net; distance to a health facility is no big problem; and proportion of low cluster wealth status. These were derived from the household characteristics. The classification into low and high was based on the cut-off point of 50th percentile (median) score for each variable [47].

#### 2.5.5. Area-Related Va31riables

Region of residence are the six geopolitical zones in Nigeria where the child resides in. There is seasonal variation across the zones, and this can impact differently on health outcome particularly on children under-five years; place of residence is the type of place where the child’s household is located and is classified as either urban or rural; state human development index (SHDI). The human development index by each state defines the average on key indicators of human development of the communities. The higher the HDI, the more protective the children from these communities will be from diseases. These categorizations of HDI by states in this study as given in NHDR 2018 [28]; state multidimensional poverty index (SMPI), is the measure of the socioeconomic status of the state and is taken as proxies to the multidimensional poverty index of the communities. The highly deprived a community’s MPI is, the more harmful the people are to diseases. This categorizations of HDI by states are also given in NHDR 2018 [28].

### 2.6. Statistical Analysis

Descriptive analysis using percentage frequencies was used to establish the prevalence, distribution, and association of the malaria status among children 6–59 months of age in Nigeria with the predictor variables considered in this study. “*Svyset*” command was used to adjust for under- and over-reporting in the survey using a weighting factor of (v005/1000000), where v005 is the sample weight [45,48]. Given the complex/hierarchical nature of the data sets, such that children/parental/household in individual units (at Level-1), since children from the same parent, and household tend to be more similar than children from other households because they share the same characteristics, are nested in communities/clusters (at Level-2), and nested in states (at Level-3), multiple multilevel logistic regression models were fitted to determine the predictors of malaria status among 6–59 months of age in Nigeria. A likelihood ratio test was carried out to establish that the three-level model was more appropriate than the two-level model (the likelihood-ratio test is LR χ^2^ = 30.21, *p* < 0.001 for Level-2 nested in Level-3).

In handling the missing values, the listwise deletion technique was applied. In the first instance, all variables having more than 20% missingness were removed, furthermore, any variable with incomplete observations, and those with responses “I don’t know” or refusal to answer the questions were deleted [45].

#### 2.6.1. Multilevel Model Description for the Three-Level Survey on Malaria Status

The dependent variable of interest is binary and follows the Bernoulli ($\pi_{ijk}$) distribution with a logit link function:(1)$$\eta_{ijk}=\beta_{0,0}^{*}+\overset{m}{\underset{a=1}{\sum }}\beta_{a,0}^{*}W_{a,ijk\;}+\overset{n}{\underset{b=1}{\sum }}\beta_{b,0}^{*}X_{b,jk}+\overset{p}{\underset{c=1}{\sum }}\beta_{c,0}^{*}Z_{k}+\varepsilon_{0,jk}+\varepsilon_{0,k}$$
where $\eta_{ijk}$ is the predicted log odds of individual child *i* (Level-1) in community (com) *j* (Level-2), and in state (sta) *k* (Level-3) having a positive RDT for malaria. $\beta_{0,0}^{*}$ represents the overall intercept (the grand mean of Level-3), $\beta_{a,0}^{*},$ $\beta_{b,0}^{*}$, and $\beta_{c,0}^{*}$ are respectively the *m*th, the *n*th, and the *p*th coefficients associated with *W* (Level-1), *X* (Level-2), and *Z* (Level-3) predictors. Further, $\varepsilon_{0,jk}$ represents the random effect of *j*th community in *k*th state, while $\varepsilon_{0,k}$ denotes the state-level random effect, with the assumption that $\varepsilon_{0,jk\;}∼N\left(0,\;\sigma_{com}^{2}\right)$ and $\varepsilon_{0,jk\;}∼N\left(0,\;\sigma_{sta}^{2}\right)$ are identical and independently distributed [49,50]. Equation (1) has a logistic transformation, with
(2)$$\eta_{ijk}=ln\left(\frac{\pi_{ijk}}{1-\pi_{ijk}}\right)$$
and it denotes the probability that an *i*th child in the *j*th community and the *k*th state will be RDT malaria fever positive.

#### 2.6.2. Model Building

In this study, five multilevel logistic models were considered. Model 1, a null model (or empty model), without any predictors. The essence is to measure the variation across the communities and the states. Model 2 included only child-related variables; Model 3, adjusted for/parental-related variables, while for Model 4, household-related variables were added to Model 3; Model 5 (full model) was derived for all the selected variables including the area-related variables. Goodness of fit was determined using Akaike’s information criteria (AIC), such that the model with the lowest AIC was chosen as the best fit [2].

#### 2.6.3. Measure of Association

The measures of association (i.e., fixed effects) were described using adjusted odds ratio (AOR) with their corresponding *p*-values and 95% confidence intervals (CIs).

#### 2.6.4. Measures of Variations

The measures of variation (i.e., random effects) were captured using intra-cluster correlation (ICC), and variance partition coefficient (VPC).

#### 2.6.5. Intraclass Correlation Coefficient (ICC)

Intraclass correlation coefficient (ICC) represents the proportion of the total variation in the model that can be accounted for by variations across the different level of clusters. In our model (three-level model), we identified two intraclass correlation coefficients: the one pertaining to children/individuals nested in community-level, and community-level groups nested in the state-level group [50,51]. Therefore:(3)$$ICC_{com}=\frac{\sigma_{com}^{2}+\sigma_{sta}^{2}}{\sigma_{com}^{2}+\sigma_{sta}^{2}+\frac{\pi^{2}}{3}}$$

$ICC_{com}$ is the correlation between two children/individuals (unit of analysis) within the same community and the same state [52,53].

However, Equation (3), in terms of the variance partition coefficient *(VPC*) differs, as it does not have corresponding interpretation, therefore:(4)$$VPC_{com}=\frac{\sigma_{com}^{2}}{\sigma_{com}^{2}+\sigma_{sta}^{2}+\frac{\pi^{2}}{3}}$$
refers to the proportion of the total variance in the same state, but different communities [52]
(5)$$ICC_{sta}=\frac{\sigma_{sta}^{2}}{\sigma_{com}^{2}+\sigma_{sta}^{2}+\frac{\pi^{2}}{3}}$$

$ICC_{sta}$ is the correlation between two children/individuals within the same state, but different in community clusters. In VPC, it refers to the proportion of the total variance that is attributable to between state-level [52].

From (3), (4), and (5), $\sigma_{com}^{2}$ is across community variance, $\sigma_{sta}^{2}$ is the across the state variance, and $\frac{\pi^{2}}{3}≃3.29$ is between children/individuals’ variance with scale factor one, and for logistic regression [49]. The values of ICCs help to establish the need for multilevel analysis as against the single-level analysis. The rule of thumb could be that when the ICC is less than 5% at the null model, hierarchical modelling may not be necessary [54].

All computations were performed in Stata 16 SE (StataCorp LP: College Station, TX, USA). In recognition of the complexity of the survey design, weight proportion as specified in Stata was used to account for over- and under-estimation. The listwise deletion, the default missing values handling technique in Stata was applied.

## 3. Results

### 3.1. Prevalence of Malaria Fever

There was a total of unweighted sample of 10,152 and weighted sample of 10,185 children 6–59 months of age in Nigeria considered in this analysis. In Figure 1, the proportion of children 6–59 months of age in Nigeria that had malaria fever positive as assessed by RDTs was 35.5% (3418/10,185), (CI: 33.9–37.1). The proportion of six states from the northern part of Nigeria, and one state from the southwest are in the highest quintile proportion group of malaria among children 6–59 months in Nigeria. Of these seven states (areas represented in deep green colour in the map), Kebbi State had the highest proportion of children 6–59 months who were malaria positive, 77.6%, (CI: 70.2–83.5), followed by Katsina State, 55.5%, (CI: 47.7–63.1). Besides, the Federal Capital Territory (FCT), Abuja had the proportion of 29.6%, (CI: 21.6–39.0), malaria positive children of 6–59 months of age. In addition, Borno, Imo, Edo, and Lagos states with proportions of 0.159 (CI: 0.11–23), 0.15 (CI: 0.10–0.23), 14.0%, (CI: 7.5–24.6), and 3.4%, (CI: 1.8–6.0), respectively are among the seven states in the lowest quintile proportion of malaria positive children of 6–59 months of age in Nigeria. Adamawa and Kaduna states have the same proportion (35%) of malaria positive children, but Adamawa state (CI: 0.26–0.45), had a wider confidence interval than Kaduna state (CI:0.28–0.43).

### 3.2. Bivariate Analysis of Proportion and Associations between Predictors and Malaria Status

Table 1 displays the descriptions of the background characteristics, the association with malaria fever status of children 6–59 months in Nigeria and presented under; child-, parental-, household-, and area-related factors, respectively. In (a) part of the table, there were more children 12–23 months of age, 23.8% (2421/10,185) among the age groups, males. In total, there were 51.2% (5216/10,185) compared to females, and 1.7% (171/10,185), were never breastfed. The prevalence of malaria fever among children 24 months and above is more than the national prevalence of 35.5%. Furthermore, the result shows that malaria status was strongly associated with child’s, age, birth order, vaccination, malaria status, nutritional status, fever, diarrhoea, duration of breastfeeding, deworming, iron pills/syrup, and vitamin A intake.

However, malaria status was not statistically associated with child’s sex, perceived birth size, wasting, and the presence of acute respiratory infection two weeks before the survey.

The results in (b) part show the relationship between parental characteristics and malaria status of children 6–59 months of age in Nigeria, only marital status and paternal occupational status were not strongly associated with the malaria status at 5% level of significance. Whereas child’s place of delivery; preceding birth interval, maternal religious status, age group, age at first birth, educational status, working status, body mass index, anaemia status, autonomy level, ante-natal care visit, maternal ethnicity, religious status, maternal iron supplement during pregnancy, and paternal education status were all strongly associated with the child’s malaria fever status. The proportion of children with malaria fever (38%) was highest among mothers in the youngest age groups (15–24 years) compared to other age groups. More than 50% of children whose mother and father had no education reported RDT positive results.

Additionally, 38%, 38%, 36%, and 32% of children whose mother had a preceding birth interval of 12–24 months, 25–35 months, 36–59 months, and 60+ months were RDT positive, respectively. In addition, the prevalence of malaria fever among children aged 6–59 months was higher for those whose mothers were underweight compared to normal and overweight mothers.

On the household-related categories of predictors, (i.e. the (c) part of Table 1), indicates that household socioeconomic status (wealth index), household size, number of bedrooms available, number of under-five in the household, age and sex of household head, number of under-five who slept under bed net the night before the survey, the various household characteristics were statistically associated with RDT positive status among children 6–59 months of age in Nigeria at 5% level of significance, while disposal of youngest child’s stool methods and household sharing toilet facilities were not statistically associated with malaria status. The proportion of malaria fever among children 6–59 months in Nigeria varied inversely with the level of household wealth index. The highest was recorded among the poor household (53%) followed by middle (38%) and rich (18%) household.

Accordingly, the household where only some under-five years slept under a bed net the night before the survey witnessed the highest prevalence of malaria fever among children 6–59 months of age in Nigeria when compared to the household where “no net in the house” (30%), “no child” (34%), “all children” (37%), slept under a bed net the night before the survey.

In addition, (d) of Table 1 displays the results of the univariate analysis and the association between the cluster-related factors and the malaria fever status of children 6–59 months of age in Nigeria. All the three variables were strongly associated with malaria fever status of children 6–59 months in Nigeria.

Furthermore, (e) of Table 1 displays the results of the univariate analysis and the association between the area-related factors and the malaria fever status of children 6–59 months of age in Nigeria. All the area variables (human development index, multidimensional poverty index, regions of residence, and place of residence) were strongly associated with malaria fever status of children 6–59 months in Nigeria.

### 3.3. Multilevel Multivariable Models of Predictors of Malaria Fever Status

In the first instance, all the variables that serves as proxies to nutritional status and household wealth were excluded from the multilevel analysis. Furthermore, a multicollinearity test was conducted to check for highly correlated predictors. Out of the variables included two factors: “under-five slept under a bed net the night before the survey” and “household had bed net” were perfectly correlated with variance inflation factors (VIF) of 7.08 and 11.18, respectively such that the mean VIF was 2.23. The variable “household had bed net” was dropped resulting in a mean VIF of 1.79. We used a forward stepwise variable selection procedure by entering all variables that were statistically associated with the malaria status of children 6–59 months of age in Nigeria at a 5% level of significance, and removal was by *p* > 0.20. Because of this, 25 variables (child’s age, duration of breastfeeding, anaemia status, nutritional status, fever status, deworming, maternal age in group, age at first birth, maternal education status, paternal education status, maternal anaemia status, ethnic group, religious status, household wealth, number of under-five in household, household head age group, under-five slept under a bed net, number of bedrooms, low cluster wealth level, cluster distance to health facility is no big problem, low cluster household with bed net, state multidimensional poverty index, state human development index, region of residence, and place of residence), were finally retained for the multilevel model building.

#### 3.3.1. Multilevel Model Results

##### A Measure of Variation (Random Effects)

Model 1 is the null model (no predictors) with the fixed effect showing that the proportions of the total variations due to differences in the communities and the states were respectively, 1.266 and 0.614, while the variance due to individual level is 3.29 (π^2^/3), which is fixed for logit.

Therefore, the variations in the prevalence of malaria status due to the three-level factors were assessed through intrastate correlation coefficient of 0.1188 (95% CI: 0.75–0.183) and intracommunity correlation coefficient of 0.3636 (95% CI: 0.318–0.412), indicating that 11.9% and 36.4% of the total variation in the odds of malaria positive were respectively due to state and community levels. The variance partition coefficient (VPC) at the state level corresponds with the ICC at the state level. However, the VPC at community level is 0.249, meaning that 24.9% of the total variance is collectively attributed to both the state and community levels. However, from the chosen model (Model 4) in Table 2, the ICC at the community-level has dropped from 36.4% in the null model to 21.0% (95% CI:17–25%), meaning the correlation between two children/individuals (unit of analysis) within the same community and the same state is 0.21, and the ICC at state-level dropped from 11.88% to 4.8% (95% CI: 3–8%), both had remained significant. The performance of models was established using AIC and likelihood ratio. Improvements in model fit was achieved at Model 4 (full model), with AIC = 9646, and log likelihood = −4763.47.

##### Measures of Association (Fixed Effects)

Table 2 shows the results of the adjusted odds ratios (AOR) for each of the variables considered in the analysis after adjusting for the rest variables. Model 2 represents the model fitting with child/individual-level variables only: age of the child, the child’s duration of breast feeding, had fever two weeks before the survey, dewormed in the last six months before the survey, anaemic status of the child, maternal age at first birth, secondary education and above of maternal and paternal, mother’s anaemia and ethnicgroup, household wealth, number of under-five in the house is more than four children, and the household head is between 35 and 44 years of age were statistically significant predictors of malaria status among children 6–59 months of age in Nigeria. The child’s nutritional status, maternal age and religious status, number of under-five who slept under a bed net the night before the survey, and the number of bedrooms in the household were not statistically significant predictors. However, after including all the predictor variables (Model 4), the significant status of the child-specific factors remains.

The odds of a child having malaria increased as the child’s age increased. The odds of children between the age of 48–59 months experiencing malaria fever were 2.68 times the odds of children 6–11 months of age (AOR = 2.68, 95% CI: 2.03–3.54). Children who were still breastfeeding (AOR = 0.61, 95% CI: 0.51–0.76), and dewormed (AOR = 0.75, 95% CI: 0.65–0.87), had 39% and 25% reduced odds of contracting malaria infection. Similarly, children whose mother had their first birth after the age of 20 years had reduced odds of experiencing malaria fever than their counterparts whose mothers had their first birth earlier than 20 years.

In addition, children whose mothers (AOR = 0.67, 95% CI: 0.55–0.82), or fathers (AOR = 0.79, 95% CI: 0.66–0.95) had secondary education and above had 33% and 21% significantly reduced odds, respectively, of being malaria fever positive. The wealthier the household (AOR = 0.61, 95% CI: 0.49–0.76), the less likely the child can be malaria positive. From among the clusters-related variables, children from a community with high proportion of mothers who said distance to the nearest health centre is “no big problem” had lower odds of malaria fever infested. The result from among the area-specific variables shows that children from south–south geopolitical zone (AOR = 0.50, 95% CI: 0.25–0.98), had 50% reduced odds to contract malaria fever when compared with children from the north central geopolitical zone. On the contrary, children from rural area (AOR = 2.12, 95% CI: 1.75–2.57), were more likely to suffer from malaria infection compared to children from urban area.

Finally, the result shows that the state-level multidimensional poverty index and state human development index were not statistically significant predictors of malaria status among children 6–59 months of age in Nigeria.

## 4. Discussion

This study assessed the prevalence of malaria across the states and the federal capital territory and examined its predictors among children 6–59 months of age in Nigeria. Some researchers use either rapid diagnostic test or microscopy thick blood smears result to classify the presence or absence of *Pf*. However, Azikiwe et al. (2012) found that RDTs and microscopy laboratory investigation of malaria *Pf* yield similar results with RDTs being more precise [55]. In addition, in a recent scoping review [1], RDT was found to be more frequently used in studies compared to microscopic smears. Additionally, in 2018 NDHS, RDTs account for more samples than microscopic blood smear laboratory test. RDT result was used in this study. The study compliments the findings in previous studies [15,22], to show impacts of some potential effects of contextual variables at both cluster and state levels. About one-third of children were found to be malaria positive. This result shows that substantial reduction in malaria prevalence among children 6–59 months of age in Nigeria have been achieved since 2010 when the first national representative survey was conducted, (52%), and 2015 recorded 45% prevalence [9,10]. Furthermore, the study established that malaria status among children 6–59 months of age in Nigeria is determined by both child-, parental/household-, community-, and state-related variables. The results revealed that children in the higher age group are more prone to malaria infection than their younger counterparts, this agrees with other studies [5,15,16,21]. The reason for this may be that the younger the child, the more attention he/she gets from caregivers, i.e., priority is often given to the younger ones in the use of bed net when available. However, children less than 12 months may not have started walking, and as the age increases and the likelihood to walk it becomes more difficult to protect them. The study also discovered that currently breastfeeding children had reduced odds of malaria infections of which this agrees with another study [13]. The possible reasons for these are that a breastfeeding child gets more antibodies from the mother to help fight infection, and a breastfeeding child often sleeps alongside the mother who normally ensures that the child is unexposed to mosquito bites and this could provide additional protection [5]. Furthermore, there are evidence that comorbidity in anaemia and non-malaria fever increased the chance of malaria infection [5] but had decreased influence with anaemia [13].

As the maternal age at first birth increases, the odd of child’s RDT outcome decreases. The study also found that maternal and paternal education statuses are significantly associated with child’s malaria status. The odds reduced with an increase in educational level [5,13,21]. However, it is not clear how these two factors interact to affect child’s health outcomes, (this can be interesting for future analysis). This finding further demonstrates the important role parental education plays in a child’s health [5]. In addition, the result shows that the anaemia status of the mother is significantly associated with the child’s malaria prevalence.

Household wealth plays a significant role in child’s health, children from rich household are less likely to be infected with the malaria parasite. Similar conclusions were reached in recent studies [5,13,15,56]. Furthermore, the study revealed that the higher the number of rooms available in the household for sleep, the less likely the child will be malaria RDT positive, this agrees with the result found in a similar study [19]. This may relate to the fact that when fewer people sleep in a room, the tendency for reduced cross-infection. Children from rural areas were found to be more vulnerable to malaria infection compared to their urban counterparts, this finding is consistent with similar studies [13,14,15,16,17]. The study also shows that the higher the proportion of respondents whose distance getting to health centre “is no big problem”, the less likely is the children would contract malaria infections. In other words, when immediate action for prompt medical attention becomes a big problem in a community, the children are highly exposed to be bitten by an infected mosquito and other childhood diseases.

### 4.1. Strengths and Limitations of the Study

A recent scoping review conducted has revealed that there are very few studies that analysed the influence of contextual factors on the risk of contracting malaria infection among children under-five years in SSA countries [1]. This study has contributed to bridging this knowledge gap. Moreover, the few studies available adopted a two-level multilevel analysis procedure, but in this study, we found that a three-level multilevel analysis was more appropriate, as such, this study is one among the few to carry out such analysis at a country level. The study involved large data sets, which might make it possible to draw inference over the country.

However, there were some limitations: (i) The data sets were cross-sectional and could not ascertain the remote causes of RDT malaria positive among children 6–59 months of age in Nigeria. Information regarding the causes is better obtained from a longitudinal study which requires periodic follow up of participants [41,48]. (ii) The accuracy of the information provided at the survey time were not ascertained to be correct because of high maternal illiteracy in Nigeria which could have resulted in recall errors while responding to some questions. (iii) some variables were dropped for missingness, and listwise deletion method was used to handle incomplete data in the remaining variables. Other methods such as multiple imputation could have been used for the incomplete observation. (iv) The classification of the determinants of RDT malaria prevalence among children 6–59 months of age were tailored through child-, parental- household, and contextual framework. However, it is argued that some variables may not directly increase or decrease the risk of contracting malaria (indirect), and some may do (direct), and the interactions effects may be interesting. This was not performed in this study and could be a subject of future study.

### 4.2. Policy Implications

Between 2010 when a national representative survey on malaria was first conducted to determine the prevalence of malaria among children 6–59 months of age in Nigeria, and 2018 when the current survey was conducted, the prevalence of malaria had dropped significantly by 15% from 50% in 2010 to 35% in 2018. This drop is commendable but was not enough to meet the expectations of total elimination of malaria-related death in Nigeria by the year 2020 [6,9]. Despite the achievement obtained so far, Nigeria remains the most malaria-endemic nation in the world. In view of this, the current efforts regarding the Government, need a boost to ensure that the scare resources are channelled to areas that will require more and urgent attention. The findings from this study will aid informed decisions that will help reduce the incidence of malaria among children 6–59 months of age in Nigeria.

In line with some findings in this study, we have proffered some suggestions for policy implementation.(i)The findings show that older children are often at risk compared to younger siblings. These children are allowed by their parents to walk about freely unattended to, perhaps to obtain more feeding opportunities elsewhere. These children have been weaned from breast feeding, nutritious foods are often not provided at home because some households cannot afford it. Most of these children scavenge about to find food. It is recommended that the school feeding program of the Government should also be extended to what may be called “community feeding program” where children of pre-school age can access food and boost their immunity.(ii)Parenting is a very serious matter in our society. The age at which most women first become a mother is often very low, thereby making them “baby-mothers”. Some of these mothers are not even mature enough to take care of themselves, particularly becoming mothers at that tender age, and not even able to care of their babies. Policies and the political will should be in place to discourage early girl-child marriage that often result into early childbirth. Girl-child education often delay the age at which most females become married and give birth. Therefore, female education should be made free and compulsory for up to secondary education. Any parent who withdraws their girl child from school for early marriage or whose daughter gets pregnant while still in school should be liable for prosecution.(iii)Most families sleep in over-crowded apartments. The governmental housing policies in some states see some buildings constructed but are never allotted to anyone until they are eventually vandalized. The Government’s policies should encourage early allocation of these buildings to those that need them to ease off over-crowding households and communities. This will reduce cross-infections in communicable diseases such as malaria(iv)Due to the problem of accessibility to most rural areas (especially during the raining season when malaria infection rates are usually very high), these communities are often neglected in the distribution of scare palliatives from the Government and agencies (such as medicals and bed nets), to addressing the issue of malaria. The Government can also invest in the use technologies available for logistics such as drone to transport these items to the hard-to-reach communities.(v)When community distance to health facility is not a “big problem” to the people, they become prompt in getting the much-needed timely medical attention for their children. Therefore, having identified areas of high malaria prevalence, the governments for these areas should among other suggestions ensure that increased proportion of the people do not travel long distances before they can access prompt medical attention. Therefore, more functional health centres are available in such localities.(vi)Lastly, the results also indicates that children from a high category among low cluster wealth areas had significantly higher proportion of malaria positive status compared to the proportion among the low category ((d) in Table 1). Therefore, the Government can investigate and implement ways to bring many homes out of poverty lines. Many of the governmental programs in the past toward poverty alleviation have ended up without achieving their aims. These measures do not find their way into the hands of those that need them. Sometimes they are distributed as political campaign “juices” for political party supporters. There should be sincerity in the part of both the program implementers and the beneficiaries in driving the program to success.

## 5. Conclusions

Despite the huge resources committed to eliminating malaria induced morbidities and mortalities, Nigeria has remained the most burdened malaria-endemic nation in the world. This study has identified some important individual and contextual predictors of malaria among children 6–59 months of age in Nigeria. These predictors are areas that need to be considered for policy designs and implementations toward control and total elimination of malaria-related morbidity and mortality among children in Nigeria.

## Figures and Tables

**Figure 1 ijerph-18-11234-f001:**
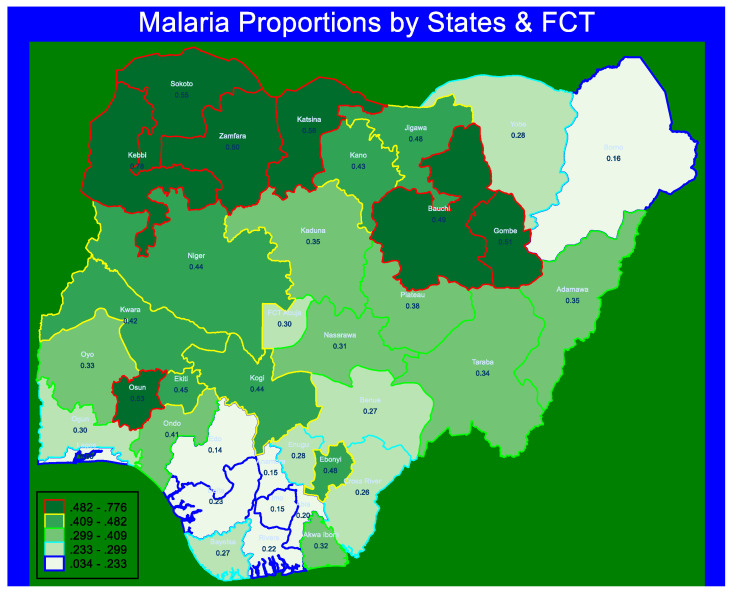
Spatial plot of the proportion of malaria positive children 6–59 months of age in Nigeria by states and FCT.

**Table 1 ijerph-18-11234-t001:** (**a**) Crosstabulation of malaria status versus child-related predictors. (**b**) Crosstabulation of malaria status versus parental-related predictors. (**c**) Crosstabulation of malaria status versus household-related predictors. (**d**) Crosstabulation of malaria status versus community-related predictors. (**e**) Crosstabulation of malaria status versus area-related predictors.

(a)			
Variables (Categories)	Malaria Status (6–59 Months)		
	Total	No	Yes
	(*n*)	*n* (%)	*n* (%)
**Age of the child**	10,185	χ^2^ = 148.15, *p*-value < 0.0001	
06–11 months	1232	925 (0.75)	307 (0.25)
12–23 months	2421	1686 (0.70)	736 (0.30)
24–35 months	2159	1379 (0.64)	780 (0.36)
36–47 months	2229	1332 (0.60)	896 (0.40)
48–59 months	2143	1245 (0.58)	898 (0.42)
**Sex**	10,185	χ^2^ = 0.551, *p*-value = 0.516	
Male	5216	3346 (0.64)	1871 (0.36)
Female	4968	3221 (0.65)	1747 (0.35)
**Perceived birth size**	10,060	χ^2^ = 8.006, *p*-value = 0.073	
Large	923	631 (0.68)	292 (0.32)
Average	7915	5094 (0.64)	2822 (0.36)
Small	1222	764 (0.63)	457 (0.37)
**Birth order**	10,185	χ^2^ = 157.98, *p*-value < 0.0001	
1st order	1945	1373 (70)	582 (30)
2nd or 3rd order	3483	2388 (69)	1095 (31)
4th–6th order	3207	2007 (63)	1200 (37)
7th+ order	1549	809 (52)	740 (48)
**Duration of breastfeeding**	10,185	χ^2^ = 47.86, *p*-value < 0.0001	
ever, but not currently	7441	4662 (63)	2780 (37)
never breastfed	171	102 (60)	69 (40)
still breastfeeding	2572	1803 (70)	769 (30)
**Child had diarrheal in the last 2 weeks**	10,182	χ^2^ = 41.25, *p*-value < 0.0001	
No	8832	5801 (66)	3031 (34)
Yes	1350	765 (57)	585 (43)
**Child had fever in the last 2 weeks**	10,182	χ^2^ = 304.96, *p*-value < 0.0001	
No	7487	5201 (69)	2285 (31)
Yes	2695	1366 (51)	1330 (49)
**Child had acute respiratory infections in past 2 weeks**	10,183	χ^2^ = 0.340, *p*-value = 0.6332	
No	9575	6168	3407
Yes	608	398	209
**Child took vitamin A supplements**	10,141	χ^2^ = 173.345, *p*-value < 0.0001	
No	5309	3106 (58)	2203 (42)
Yes	4832	3432 (71)	1399 (29)
**Child had deworming treatment in the last 6 months**	10,133	χ^2^ = 235.73, *p*-value < 0.0001	
No	7235	4330 (60)	2905 (40)
Yes	2898	2203 (76)	695 (24)
**Child took iron supplements**	10,151	χ^2^ = 42.173, *p*-value < 0.0001	
No	8224	5183 (63)	3040 (37)
Yes	1927	1367 (71)	561 (29)
**Nutritional status**	10,185	χ^2^ = 267.41, *p*-value < 0.0001	
Well-nourished	5688	4060 (71)	1627 (29)
Poorly nourished	4497	2507 (56)	1990 (44)
**Stunting**	10,185	χ^2^ = 297.16, *p*-value < 0.0001	
No	6285	4457 (71)	1827 (29)
Yes	3900	2109 (54)	1790 (46)
**Wasting**	10,185	χ^2^ = 4.03, *p*-value =0.1169	
No	9481	6138 (65)	3343 (35)
Yes	703	429 (61)	274 (39)
**Underweight**	10,185	χ^2^ = 156.20, *p*-value < 0.0001	
No	7915	5355 (68)	2560 (32)
Yes	2269	1212 (53)	1058 (47)
**Overweight**	10,185	χ^2^ = 6.20, *p*-value = 0.0337	
No	10,019	6445 (64)	3573 (36)
Yes	166	122 (74)	44 (26)
**Anaemia status**	10,183	χ^2^ = 649.60, *p*-value < 0.0001	
No	3241	2664 (82)	577 (18)
Yes	6942	3902 (56)	3040 (44)
**(b)**			
**Variables (Categories)** **Categories**	**Malaria Status**		
	**Total**	**No**	**Yes**
	**(*n*)**	** *n* ** **(%)**	** *n* ** **(%)**
**Maternal age (years) group**	10,185	χ^2^ = 14.59, *p*-value = 0.0095	
15–24 years	2048	1265 (62)	784 (38)
25–34 years	5262	3481 (66)	1781 (34)
35 years+	2874	1821 (63)	1052 (37)
**Maternal age at first birth**	10,185	χ^2^ = 385.04, *p*-value < 0.0001	
10–19 years	5406	3033 (56)	2373 (44)
20–29 years	4369	3177 (73)	1192 (27)
30+ years	409	357 (87)	53 (13)
**Mother working status**	10,185	χ^2^ = 8.83, *p*-value = 0.038	
Not working	2978	1855 (62)	1123 (38)
Working	7207	4712 (65)	2494 (35)
**Mother’s educational status**	10,185	χ^2^ = 864.57, *p*-value < 0.0001	
No education	3970	1951 (49)	2018 (51)
Primary education	1643	985 (60)	658 (40)
Secondary and above	4571	3631 (79)	940 (21)
**Marital status**	10,185	χ^2^ = 1.756, *p*-value = 0.5012	
Never in union	171	110 (64)	61 (36)
In union	9733	6265 (64)	3468 (36)
Widow/divorced/separated	281	191 (68)	89 (32)
**Paternal educational status**	9604	χ^2^ = 672.98, *p*-value < 0.0001	
No education	2872	1369 (48)	1503 (52)
Primary education	1423	817 (57)	606 (43)
Secondary education	5308	4018 (76)	1290 (24)
**Paternal occupation**	10,185	χ^2^ = 0.294, *p*-value = 0.7133	
Not working	304	191 (63)	112 (37)
Working	9881	6376 (65)	3505 (35)
**Mother lives with a partner**	9733	χ^2^ = 7.433, *p*-value =0.0291	
Living with partner	8862	5668 (64)	3194 (36)
Living alone	871	598 (69)	273 (31)
**Mother slept under mosquito net**	10,185	χ^2^ = 91.02, *p*-value < 0.0001	
No	4671	3242 (69)	1429 (31)
Yes	5514	3325 (60)	2188 (40)
**Mother’s body weight status**	8690	χ^2^ = 250.16, *p*-value < 0.0001	
Underweight	5310	3219 (61)	2092 (39)
Healthy	884	513 (58)	371 (42)
Overweight and obese	2537	1977 (78)	559 (22)
**Preceding birth interval**	8220	χ^2^ = 15.683, *p*-value= 0.0155	
08–24 months	2190	1356 (62)	834 (38)
25–35 months	2884	1775 (62)	1109 (38)
36–59 months	2351	1515 (64)	836 (36)
60+ months	795	544 (68)	251 (32)
**Mother’s anaemia status**	10,053	χ^2^ = 120.013, *p*-value < 0.0001	
Normal	4206	2991 (71)	1214 (29)
Anaemic	5847	3519 (60)	2328 (40)
**Number of ANC attendance**	6375	χ^2^ = 185.99, *p*-value < 0.0001	
None	1342	715 (53)	627 (47)
Less WHO number	954	624 (65)	329 (35)
Met WHO number	4079	2987 (73)	1092 (27)
**Maternal autonomy**	10,185	χ^2^ = 178.05, *p*-value < 0.0001	
Less autonomy	5071	2947 (58)	2124 (42)
more autonomy	5114	3620 (71)	1494 (29)
**Maternal ethnicity**	10,185	χ^2^ = 325.93, *p*-value < 0.0001	
Hausa/Fulani	4067	2226 (55)	1841 (45)
Ibos	1650	1273 (77)	377 (23)
Yoruba	1490	1068 (72)	421 (28)
Others	2978	2000 (67)	977 (33)
**Religion status**	10,185	χ^2^ = 255.02, *p*-value < 0.0001	
Catholic	1027	754 (73)	273 (27)
Other Christian	3438	2509 (73)	929 (27)
Islam	5655	3266 (58)	2389 (42)
Others (traditional)	64	39 (60)	25 (40)
**Place of delivery**	10,184	χ^2^ = 455.18, *p*-value < 0.0001	
Home	5348	2953 (55)	2394 (45)
Public health facility	2977	2137 (72)	840 (28)
Private health facility	1660	1334 (80)	326 (20)
Somewhere else	200	142 (71)	58 (29)
**(c)**			
**Variables (Categories)**	**Malaria Status (6–59 Months)**		
	**Total**	**No**	**Yes**
	** *n* **	** *n* ** **(%)**	** *n* ** **(%)**
**Household wealth status**	10,185	χ^2^ = 1102.49, *p*-value < 0.0001	
Poor	3882	1813 (47)	2069 (53)
Middle	2139	1335 (62)	804 (38)
Rich	4163	3419 (82)	744 (18)
**Household had mosquito bed net**	10,185	χ^2^ = 65.709, *p*-value < 0.0001	
No	3111	2187 (70)	924 (30)
Yes	7074	4389 (62)	2693 (39)
**Household member size**	10,185	χ^2^ = 159.22, *p*-value < 0.0001	
0–3 persons	980	678 (71)	282 (29)
4–6 persons	4835	3322 (69)	1513 (31)
7–9 persons	2461	1521 (62)	940 (38)
10+ persons	1908	1026 (54)	881 (46)
**Number of bedrooms in household**	10,185	χ^2^ = 47.584, *p*-value < 0.0001	
One room	2807	1952 (70)	854 (30)
Two rooms	3489	2221 (64)	1268 (36)
Three rooms	2030	1239 (61)	791 (39)
Four rooms	981	604 (62)	377 (38)
Five+ rooms	877	550 (63)	326 (37)
**Number of children under-five in household**	10,185	χ^2^ = 130.40, *p*-value < 0.0001	
No children or one child	2700	1880 (70)	819 (30)
Two children	4315	2848 (66)	1468 (34)
Three children	2054	1270 (62)	783 (38)
Four children+	1115	568(51)	547 (49)
**Improved source of drinking water**	10,185	χ^2^ = 298.76, *p*-value < 0.0001	
Unimproved	3078	1601 (52)	1477 (48)
Improved	7106	4966 (70)	2140 (30)
**Improved toilet facilities**	10,185	χ^2^ = 650.37, *p*-value < 0.0001	
Unimproved	4607	2357 (51)	2250 (49)
Improved	5577	4210 (75)	1367 (25)
**Youngest child’s stool disposal**	6408	χ^2^ = 0.102, *p*-value= 0.8208	
Proper	3606	2348 (65)	1257 (35)
improper	2803	1836 (66)	967 (34)
**Improved floor material type**	10,185	χ^2^ = 329.83, *p*-value < 0.0001	
Unimproved	2877	1460 (51)	1418 (49)
Improved	7307	5107 (70)	2200 (30)
**Improved roofing materials**	10,185	χ^2^ = 87.795, *p*-value < 0.0001	
Unimproved	1125	583 (52)	542 (48)
Improved	9060	5984 (66)	3076 (34)
**Improved wall materials**	10,184	χ^2^ = 638.88, *p*-value < 0.0001	
Unimproved	3265	1535 (47)	1730 (53)
Improved	6919	5032 (73)	1887 (27)
**Sex of household head**	10,185	χ^2^ = 7.815, *p*-value= 0.0283	
Male	9008	5824 (64)	3273 (36)
Female	1087	743 (68)	344 (32)
**Household head age (years) group**	10,185	χ^2^ = 27.139, *p*-value= 0.0026	
less 34 years	2828	1825 (65)	1003 (35)
35–44 years	3946	2648 (67)	1298 (33)
45–55 years	2091	1300 (62)	792 (38)
56 years+	1318	794 (60)	524 (40)
**Household had electricity**	10,066	χ^2^ = 590.89, *p*-value < 0.0001	
No	4296	2186 (51)	2109 (49)
Yes	5771	4296 (74)	1475 (26)
**Type of cooking fuel**	10,182	χ^2^ = 384.85, *p*-value < 0.0001	
Electricity and Gas	1211	1088 (90)	123 (19)
Biofuel/mass	8971	5477 (61)	3494 (39)
**Under-five slept under bed net**	10,112	χ^2^ = 104.81, *p*-value < 0.0001	
No child	1317	863 (66)	454 (34)
All children	4715	2965 (63)	1750 (37)
Some children	996	530 (53)	466 (47)
No net in the house	3083	2165 (70)	918 (30)
**(d)**			
**Variables (Categories)**	**Malaria Status (6–59 Months)**		
	**Total**	**No**	**Yes**
	** *n* **	** *n* ** **(%)**	** *n* ** **(%)**
**Proportion of low cluster wealth level**	10,185	χ^2^ = 842.35, *p*-value < 0.0001	
Low	5323	4133 (78)	1189 (22)
High	4861	2433 (50)	2428 (50)
**Distance to health facility is no big problem**	10,185	χ^2^ = 233.12, *p*-value < 0.0001	
Low	4702	2664 (57)	2038 (43)
High	5483	3903 (71)	1579 (29)
**Proportion of low cluster household with bed net**	10,185	χ^2^ = 210.75, *p*-value < 0.0001	
Low	4981	2861 (57)	2121 (43)
High	5202	3705 (71)	1497 (29)
**(e)**			
**Variables (Categories)**	**Malaria Status (6–59 Months)**		
	**Total**	**No**	**Yes**
	** *n* **	** *n* ** **(%)**	** *n* ** **(%)**
**State Human Development Index (SHDI)**	10,185	χ^2^ = 456.50, *p*-value < 0.0001	
Lowest HDI	2150	1219 (57)	839 (43)
Low HDI	2416	1297 (54)	1120 (46)
Average HDI	2223	1511 (68)	712 (32)
High HDI	2680	1886 (70)	793 (30)
Highest HDI	716	654 (91)	62 (9)
**Region of residence**	10,185	χ^2^ = 428.79, *p*-value < 0.0001	
North-central	1436	906 (63)	530 (37)
North-east	1573	1034 (66)	538 (34)
north-west	2967	1502 (51)	1465 (49)
South-east	1328	826 (76)	336 (25)
South-south	1086	826 (76)	260 (24)
South-west	1794	1307 (73)	487 (27)
**State Multidimensional Poverty Index (SMPI)**	10,185	χ^2^ = 364.70, *p*-value < 0.0001	
Highly Deprived	847	481 (57)	366 (43)
Above Averagely deprived	3093	1659 (54)	1434 (46)
Averagely Deprived	2318	1499 (65)	819 (35)
Mildly deprived	1939	1395 (72)	544 (28)
Lowest deprived	1987	1532 (77)	455 (23)
**Place of residence**	10,185	χ^2^ = 724.32, *p*-value < 0.0001	
Urban	4485	3538 (79)	946 (21)
Rural	5700	3029 (53)	2671 (47)

**Table 2 ijerph-18-11234-t002:** Multilevel multivariate models of predictors of malaria with adjusted odds ratios (AOR) among children 6–59 months in Nigeria.

Variables	Model 2 (*n* = 9277) (Level-1 Factors Only)			Model 3 (*n* = 9277) (Added Level-2 Factors)			Model 4 (*n* = 9277) (Added Level-3 Factors)		
Individual Level	AOR	95% CI	*p*-Value	AOR	95% CI	*p*-Value	AOR	95% CI	*p*-Value
**Child’s age**									
6–11 months	1.00			1.00			1.00		
12–23 months	1.26	1.02–1.56	0.031	1.26	1.02–1.56	0.029	1.28	1.04–1.58	0.021
24–35 months	1.65	1.26–2.16	<0.001	1.65	1.26–2.15	<0.001	1.65	1.26–2.16	<0.001
36–47 months	2.20	1.67–2.88	<0.001	2.20	1.67–2.89	<0.001	2.20	1.68–2.89	<0.001
48–59 months	2.69	2.04–3.55	<0.001	2.68	2.03–3.54	<0.001	2.66	2.02–3.51	<0.001
**Duration of breastfeeding**									
Ever breastfed	1.00			1.00			1.00		
Never breastfed	1.28	0.84–1.94	0.251	1.26	0.83–1.92	0.276	1.28	0.84–1.96	0.243
Still breastfeeding	0.63	0.51–0.76	<0.001	0.62	0.51–0.76	<0.001	0.61	0.50–0.75	<0.001
**Anaemia status**									
Not anaemic	1.00			1.00			1.00		
Anaemic	3.84	3.36–4.39	<0.001	3.82	3.34–4.37	<0.001	3.82	3.34–4.37	<0.001
**Nutrition status**									
Well-nourished	1.00			1.00			1.00		
Poorly nourished	1.07	0.95–1.2	0.284	1.06	0.94–1.19	0.378	1.05	0.94–1.19	0.386
**Fever in last 2 weeks**									
No	1.00			1.00			1.00		
Yes	1.95	1.72–2.2	<0.001	1.94	1.71–2.2	<0.001	1.96	1.73–2.22	<0.001
**Dewormed in last 2 weeks**									
No	1.00			1.00			1.00		
Yes	0.75	0.65–0.87	<0.001	0.75	0.65–0.87	<0.001	0.75	0.65–0.87	<0.001
**Maternal age (years) in group**									
15–24 years	1.00			1.00			1.00		
25–34 years	1.04	0.89–1.22	0.611	1.05	0.9–1.23	0.532	1.06	0.91–1.24	0.464
35 years+	1.16	0.95–1.4	0.143	1.17	0.96–1.42	0.111	1.19	0.98–1.44	0.085
**Maternal age at first birth**									
10–19 years	1.00			1.00			1.00		
20–29 years	0.82	0.72–0.93	0.003	0.82	0.72–0.94	0.003	0.81	0.71–0.93	0.002
30 years+	0.52	0.35–0.77	0.001	0.52	0.36–0.77	0.001	0.51	0.35–0.75	0.001
**Maternal education status**									
No education	1.00			1.00			1.00		
Primary	0.82	0.68–0.99	0.038	0.85	0.7–1.03	0.093	0.86	0.71–1.04	0.128
Secondary+	0.61	0.5–0.75	<0.001	0.65	0.53–0.79	<0.001	0.67	0.55–0.82	<0.001
**Paternal education status**									
No education	1.00			1.00			1.00		
Primary	0.87	0.71–1.06	0.157	0.89	0.73–1.08	0.244	0.90	0.74–1.10	0.304
Secondary+	0.74	0.62–0.88	0.001	0.77	0.64–0.92	0.004	0.80	0.66–0.95	0.013
**Maternal anaemia status**									
Not anaemic	1.00			1.00			1.00		
Anaemic	1.24	1.11–1.39	<0.001	1.24	1.1–1.39	<0.001	1.23	1.1–1.38	<0.001
**Maternal ethnic group**									
Hausa/Fulani/Kanuri	1.00			1.00			1.00		
Ibo	0.83	0.54–1.28	0.401	0.86	0.56–1.32	0.489	0.81	0.49–1.31	0.387
Yoruba	1.57	1.08–2.26	0.017	1.61	1.11–2.34	0.012	1.45	0.98–2.15	0.064
Others	1.36	1.08–1.71	0.010	1.33	1.05–1.68	0.016	1.29	1.02–1.63	0.037
**Maternal religion status**									
Catholics	1.00			1.00			1.00		
Other Christian	0.89	0.7–1.14	0.359	0.91	0.72–1.16	0.460	0.92	0.72–1.17	0.491
Islam	0.82	0.6–1.11	0.199	0.85	0.62–1.15	0.288	0.90	0.66–1.23	0.499
Traditionalists	0.78	0.39–1.54	0.467	0.78	0.39–1.54	0.470	0.80	0.41–1.58	0.526
**Household wealth**									
Low	1.00			1.00			1.00		
Middle	0.71	0.6–0.84	<0.001	0.84	0.7–1.01	0.070	0.86	0.71–1.03	0.102
Rich	0.43	0.36–0.52	<0.001	0.55	0.44–0.69	<0.001	0.61	0.49–0.76	<0.001
**Number of under-5 in household**									
No children or one child	1.00			1.00			1.00		
Two children	1.03	0.9–1.19	0.667	1.04	0.90–1.20	0.584	1.04	0.91–1.20	0.556
Three children	1.12	0.94–1.33	0.222	1.12	0.94–1.34	0.201	1.11	0.93–1.32	0.249
Four children+	1.48	1.18–1.85	0.001	1.47	1.17–1.84	0.001	1.46	1.16–1.83	0.001
**Household head age (years) group**									
Less 35 years	1.00			1.00			1.00		
35–44 years	0.85	0.73–0.98	0.031	0.85	0.73–0.99	0.041	0.86	0.74–1.00	0.050
45–55 years	0.87	0.72–1.05	0.135	0.89	0.73–1.07	0.204	0.90	0.75–1.09	0.270
56 years+	1.07	0.86–1.32	0.549	1.10	0.89–1.36	0.386	1.12	0.90–1.38	0.304
**Under-5 slept under a bed net**									
No child	1.00			1.00			1.00		
All children	0.89	0.74–1.07	0.217	0.89	0.74–1.07	0.202	0.88	0.73–1.06	0.176
Some children	1.15	0.91–1.46	0.244	1.16	0.91–1.47	0.234	1.15	0.91–1.46	0.239
No net in household	0.96	0.8–1.17	0.701	0.99	0.81–1.2	0.887	0.98	0.8–1.19	0.802
**Number of bedrooms in household**									
One room	1.00			1.00			1.00		
Two rooms	1.03	0.88–1.20	0.705	1.01	0.87–1.18	0.894	1.00	0.86–1.17	0.989
Three rooms	1.07	0.89–1.28	0.477	1.04	0.86–1.25	0.692	1.02	0.84–1.22	0.869
Four rooms	0.91	0.72–1.14	0.392	0.87	0.69–1.1	0.240	0.84	0.67–1.06	0.148
Five+ rooms	0.82	0.64–1.05	0.122	0.78	0.61–1.01	0.062	0.76	0.59–0.98	0.037
**Cluster level**									
**Proportion of cluster’s household with no bed net**									
Low				1.00			1.00		
High				0.92	0.76–1.12	0.410	0.97	0.80–1.17	0.718
**Distance to a health facility is no big problem**									
Low				1.00			1.00		
High				0.72	0.6–0.86	<0.001	0.76	0.64–0.90	0.002
**Proportion of low cluster wealth status**									
Low				1.00			1.00		
High				1.41	1.13–1.75	0.002	1.15	0.92–1.43	0.226
**State-level**									
**Region of residence**									
North central							1.00		
North-east							0.48	0.22–1.05	0.065
North-west							1.46	0.62–3.45	0.387
South-east							1.07	0.51–2.25	0.854
South-south							0.50	0.25–0.98	0.045
South-west							1.44	0.64–3.25	0.378
**Type of place of residence**									
Urban							1.00		
Rural							2.12	1.75–2.57	<0.001
**State human development index (HDI)**									
Lowest HDI							1.00		
Low HDI							1.32	0.71–2.45	0.374
Average HDI							1.50	0.68–3.33	0.314
High HDI							1.87	0.73–4.80	0.192
Highest HDI							1.03	0.35–2.98	0.961
**State Multidimensional poverty index (SMPI)**									
Highly deprived							1.00		
Above averagely deprived							1.75	0.90–3.43	0.101
Averagely deprived							1.51	0.62–3.68	0.362
Mildly deprived							1.20	0.44–3.26	0.721
Lowest deprived							1.29	0.41–4.03	0.665
**Intercept**	0.24	0.14–0.40	<0.001	0.20	0.11–0.35	<0.001	0.07	0.02–0.21	<0.001
**Random effect**									
Community-level variance	0.73	0.58–0.90		0.74	0.59–0.91		0.67	0.54–0.84	
State-level variance	0.38	0.22–0.65		0.39	0.23–0.68		0.20	0.11–0.36	
VPC: child-level	0.749			0.74			0.79		
VPC: community-level	0.165			0.167			0.161		
VPC: state-level	0.09			0.089			0.048		
ICC%: community-level	25.18	21–30		25.57	21–30		21.00	17–25	
ICC%: state-level	8.65	5–14		8.90	5–14		4.82	3–8	
**Model fit statistics**									
Log-likelihood	−4818.16			−4804.56			−4763.47		
AIC	9722.32			9701.12			9646.94		
BIC	10,029.14			10,029.35			10,075.06		

AOR: adjusted odds ratios, ICC: intraclass correlation coefficient, VPC: variance partition coefficient, AIC: Akaike information criterion (given a set of candidate models for the data, the preferred model is the one with the minimum AIC value), BIC: Bayesian information criterion.

## Data Availability

The data set used in this study is available in MeasureDHS https://dhsprogram.com (accessed on 28 January 2020) and UNDP-Nigeria http://hdr.undp.org/sites/default/files/hdr_2018_nigeria_finalfinalx3.pdf (accessed on 3 March 2020).

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
