# Peer review of "Individual and Contextual Factors Associated with Malaria among Children 6–59 Months in Nigeria: A Multilevel Mixed Effect Logistic Model Approach"

_ijerph, 2021, doi:10.3390/ijerph182111234_

Round 1
Reviewer 1 Report
Individual and contextual factors associated with ma-laria among children 6-59 months in Nigeria: A multi-level mixed effect logistic model approach, by Phillips Edomwonyi Obasohan et al.,
This work comprises”data from 2018 Nigeria Demographic and Health Survey (NDHS) involving a weighted sample size of 10,185 children who were tested for malaria using rapid diagnostic test (RDT)“,“had malaria fever positive“ and “ multilevel mixed effect logistic regression models were used .
The two studied states had high positivity of the RDT test; 55 and 77%. In the abstract, authors mentioned only two outstanding asscociations with malaria fever; older children U5 and living in a rural area. There are previous publications that study similar parameters, however it is confusing why the references used to cite them are different in the Introduction and discussion sections.
On page 2: Introduction: “This study enlarges the findings in previous studies to show proofs of contextual variables at both community and state levels [17,18]. Therefore, this study aimed to “establish the prevalence of malaria across the states and federal capital territory, and to ex-amine the individual- and contextual-level predictors of malaria fever among children 6-59 months of age in Nigeria.“ and on page 19; Discussion: “ The study enlarges the findings in previous studies [4,10,35],”. I have a concern about the references cited in the introduction versus the discussion when authors mentioned previous studies similar to this manuscript.
Suggestions and comments:
The manuscript is well-written and understandable, however, previous studies have shown that young children U5 and rural areas are more associated to malaria positivity. I suggest, if possible to include risks maps of children U5 malaria positive at community and state levels, that can be useful for policy makers.
Methods/Results. Page 3: “(ii) Laboratory microscopy investigation on thick blood smears was done for a three quarter of the households where RDTs was done” please, include what was the correlation of both tests. There is no distinction between being positive to RDT test as first time or having previous infections, as well as if all children had an uncomplicated disease.
Different independent variables were included in the descriptive analysis (section 2.5). However, it is useful to separate the direct vs indirect variables (also the predictors indicated in tables) , and might linked indirect independent variables to a direct one. Because might be a bit confusing, to evaluate some variables that have litle to do with acquiring the malaria, e.g. child nutrition, breast feeding, as other individual factors might play a role in preventing a severe disease, but not lo acquire malaria.
There are parameters indirect independent variables as duration of breastfeeding, Anaemia status, Nutrition status, Dewormed in last 2 weeks, Maternal age group, etc that might be associated to percepción of “good health” and therefore these people take care of their children from any health issues, in general, including malaria.
Please clarify what is the streighten of three-level multilevel analysis over two-level multilevel analysis:“we found that a three-level multilevel analysis was more appropriate, as such the study is the first 59 to carry out such analysis in Nigeria”
In the Discussion some arguments should be revised, the odds for malaria positivity considering indirect independent variables e.g. currently breastfeeding children had reduced odds of malaria. Page 19, lines 22-24. I suggest to consider that children <11 months do not walk, and as age increases they do start walking and is more difficult to protect them.
“The possible reason for this is that a breastfeeding child often sleeps alongside the mother who normally would ensure that the child is unexposed to mosquito bites and the maternal antibodies could provide additional protection [4].” Please revise the sentence about -additional protection-. Because the work, in general, refers to protection from mosquito bite.
Last line 51, previuosly -page 2/22-; seems incomplete “This is consistent with the findings in”
The following statement agreed with my previous comment, if families live nearby a health facility they are exposed to information on preventive measures tan might help them to prevent mosquito bites: Page 19 “The study also shows that the higher the proportion of respondents whose distance getting to health centre ‘is no big problem’, the less likely is the children contract malaria infections. In other words, when getting to health facility for prompt medical attention becomes a big problem in a community, the children are highly exposed to the risk of malaria fever and other childhood disease”
Please revise other minor issues, e.g.
On page 3: “[26] found that RDTs and microscopy laboratory investigation of malaria Pf yield similar results with RDTs being more precise [26].” Ref [26] is repeated.
4.1 section: “Also, in a recent scoping re-view (Obasohan et al 2021), RDT” might be better to include the number of the reference.
Author Response
We deeply appreciate your constructive comments on our work which has definitely improved the quality of work done
We have submitted herein the responses to some of the issues raised in your review

Reviewer 2 Report
I have noticed some elements that should be corrected by the authors:
- part 2.4, last pararaph, 2nd sentence: after "however", replace the number 26 by the names of the authors, and in the 3rd sentence, replace "obasohan et al 2021" by the reference number. Moreover the whole paragraph should be moved to the discussion part of the manuscript.
- part 3.2, line 6, the meaning of the percentage 1.7% is not specified.
Author Response
We want to say thank you for finding time to review our paper, and for your kind comments and constructive suggestion which has increased the quality of the paper.
We submit herein the responses to some of the issues raised in your review

Reviewer 3 Report
The authors set out to understand individual and contextual risk factors among children under 5 years of age in Nigeria. Nigeria is the most malarious country in the world, and this paper is a welcome contribution to understanding factors associated with its high prevalence. I have a number of comments for the authors, and I list them below.
Major:
Some of the statistical results are interpreted incorrectly. For example, the authors state “children from South-South geopolitical zones (AOR=0.50, 95% CI: 8 0.25-0.98), were 50% less likely to contract malaria fever than children from the North 9 central zone”. This is incorrect. The odds are 50% less, not the likelihood/probability. The authors need to go through the manuscript and change instances of this to reflect the effect on odds, not likelihood.
The authors suggest in the introduction and discussion that their work will show proof that of contextual variables influencing transmission. Proof is too strong a claim, and I recommend the authors simply say that they identify potential contextual variables associated with transmission.
The authors should contextualize their findings with previous malaria surveys conduct in Nigeria. For example, I believe there were DHS/MIS surveys done in 2010 and 2015. The results of those are published online as final reports. It would be useful if the authors discuss how prevalence may have changed over time, whether overall or by geography or demographics, etc.
Minor:
Figure 1, I would round the prevalence estimates and uncertainty intervals to 2 digits, e.g. 0.78 instead of 0.776. It makes it much easier to read.
Author Response
We appreciate you for finding time to review our paper, and for your kind comments and constructive suggestion which has increased the quality of the paper.
We submit herein the responses to some of the issues raised in your review

Reviewer 4 Report
Thanks for the opportunity to revise the manuscript “Individual and contextual factors associated with malaria among children 6-59 months in Nigeria: A multi-level mixed effect logistic model approach”. This manuscript has the goal to establish the prevalence of malaria across the states and federal capital territory, and to examine the individual- and contextual-level predictors of malaria fever among children 6-59 months of age in Nigeria. the article is interesting, well written, and does a good job at what it sets out to do. I would just suggest adding a map showing the study area, showing the EAs and the households used. I also suggest the inclusion of another map with the number of children with Malaria per study area. This second map can replace the figure 1.
Author Response
We want to say thank you for finding time to review our paper, and for your kind comments and constructive suggestion which has increased the quality of the paper.
We submit herein the responses to some of the issues raised in your review:
|
I would just suggest adding a map showing the study area, showing the EAs and the households used. I also suggest the inclusion of another map with the number of children with Malaria per study area. This second map can replace the figure 1 |
Thanks for the suggestion
Fig 1 has been replaced with risk map by states |
Reviewer 5 Report
- Please provide a persuasive reason for selecting those children between 6 and 59 months old for the study. Why not children over 60 months old?
- Please provide more details about the sample size (N=10,185). In detail, please explain how many participants are in the entire data set of NDHS, and how the number came down to the reported sample size. Also, it is unclear if 10,185 is after or before weighing. If the former, please report both numbers.
- Please explain how representative the two data sets used for analysis. Sampling weights were not used, and sampling design was not considered when it comes to analysis, so the representativeness of the data set used for analysis is questionable without more information provided.
- Please indicate the level of each group of variables throughout the text and in the tables. The variables used for analysis were at the children/individual, parental, household, community, and state levels, but it is unclear which ones were measured at levels 1, 2, and 3.
- Please explain why the household level was not considered for the multilevel analysis despite the fact that the data set provides some variables at the household level.
- It is unclear why only AIC was reported. Please report both AIC and BIC together for goodness of fit.
- Please report missing data information in the tables by adding another column next to N. And if possible, please consider missing data analysis, such as multiple imputations.
- It is noteworthy that all variables in Table 2b, except the partner’s educational status, are about the mother. Please consider changing the table title and revise some text throughout the manuscript.
- In Table 1a, 1b, 1b, and 1d, please consider bolding each variable (bot subcategories) and change the alignment as done in Table 2.
- Model 1 in Table 1 can be excluded. It is empty and already explained in the text under “A measure of variation (random effects)”.
- Please provide scientific and persuasive reasoning about the cutoff points and/or the categorizations of the following variables: child’s age group, nutrition status, maternal age group, age at first birth, number of under 5 in household, household head age group, under 5 slept under a bed net, number of rooms for sleep, cluster level proportion of cluster’s household with no bed net, distance to a health facility is no big problem, the proportion of low cluster wealth status, HDI, and MDPI. Many subjective terms were used, such as “well”, “poorly”, “some”, “low”, “high”, “highly”, “mildly”, etc. It is unclear what it means by “lowest” and “highest” HDI. It is unclear why 6-11, 12-23, 24-35, 36-47, and 48-59 age groups were used for child’s age. It is unclear why number of under 5 in household and number of rooms for sleep were considered as categorical variables, not continuous variables.
- Please discuss what the results imply. The current paragraphs in the discussion are simply describing the findings from Table 2 without discussing the implications of the main results. Especially, there is no meaningful discussion about what the results of contextual factors, at the community and state levels, actually mean to Nigerian people, which is the main focus of the study.
Author Response
We want to say a big thank you for finding time to review our paper, and for your detailed comments and constructive suggestions which have increased the quality of our work.
We submit herein the responses to some of the issues raised in your review

Round 2
Reviewer 1 Report
Individual and contextual factors associated with malaria among children 6-59 months in Nigeria: A multi-3 level mixed effect logistic model approach
By Obasohan et al.,
The authors responded most observations and added additional information that have strengthened their manuscript. However, the manuscript need a minor concerns .
In the manuscript some parts need revision, and many were highlighten, e.g. need space, revise sentence or spelling, change type of letter (capital or small case), delete a symbol, add comma, typhographical errors, etc.
other specific comments:
Table 1 ( seems new). It might be place in supplementary materials. For content in columns for Variables, Classifications and Definitions….. revise consistency. The name starts with capital letter only first word or not. I have indicated many of the inconsistentes, but not all of them.
This problem is also observed in other Tables, please revise all of them
In the text, sometimes the numbers have comma, others do not. Verify consistency.
Page 3:
What means… genotype testing? Pf drug resistance genotype testing? Or otherwise?
It is strange why drug resistance is not an issue or variable to explore as predictor of infection risk.
Page 11
Revise the following sentence, it is confusing: “ In Table 2a, there were more children between 12 and 23 months (2421/10185), males (5216/10185), and 1.7% (171/10185), never breastfed in the survey among the age group, sex, and duration of breastfeeding, respectively. “
Table 2 a
For some analysis “p-value 0.6332“ please change to p-value = 0.6332
Page 13
It was been a concern, mentioned before, that some variables might not directly increase/decrease the risk for malaria infection, and because of that would have been interesting to see how they interact (direct and indirect variables). Sorry i did not see these specific points. Sometimes, information of indirect variables might be more accesible for analysis and mirror others directly related to malaria transmission, If not analyzed, could be added to the discussion, and this is up to the authors.
“Whereas child’s place of delivery; preceding birth interval, maternal religious status, age group, age at first birth, educational status, working status, body mass index, anaemia status, autonomy level, ante-natal care visit, maternal ethnicity, religious status, maternal iron supplement during pregnancy, and paternal education status were all strongly associated with the child’s malaria fever status.“ or those mentioned in lines 400-407.

Author Response
We appreciate your constructive reviews and many suggestions that has greatly improved the manuscript.
The following table contained the responses to some points raised in your second review report

Reviewer 5 Report
- Table 1 includes rich descriptions of all the variables reported in this study. Detailed information and explanations are really appreciated, but not all variables need to be explained this much (e.g., gender). The common practice is reporting classifications with citations in text as brief as possible. Further, some classifications are still vague. For example, for “Mother’s age group”, it is unclear what “perceived experienced mothers” mean without any citation. It is unclear why mother’s age at first birth was not used as a numeric (continuous) variable. For “Household wealth status”, there is no explanation about why three categories were derived out of five and how. For “Household size”, there is no citation about the “typical” household size in Nigeria. For cluster-related variables, there is no enough explanations about how low is “low” and how high is “high”. Same issue with area-related variables. That was the issue before, and it cannot be solved by simply providing lengthy descriptions of the variables.
- There is no consensus on which one, AIC or BIC, should be used or reported, so the common practice is reporting both. If they do not agree to each other, then that is one major reason to report both.
- Please report the N for model 2, 3, and 4 in Table 4. There should be one N for all these models since they need to use the same sample. Further, this N should match the N in Table 2a, 2b, and 2c. N will decrease when partner’s (it’s “paternal” in Table 4 by the way) educational status (N=9604) is included in the model for example.
- This is related to the previous comment. The authors reported that listwise deletion is used to handle missing data, but some of the child/parent-related variables have missing data. When it comes to listwise deletion, the common practice is to use the same sample for all analysis, including descriptive statistics, final model, and anything between. This is not the case for this study. Descriptive statistics for numerous variables at level 1, 2, and 3 were reported but it is impossible to know the exact characteristics of the sample used for the final analysis since descriptive statistics reported include all available data, not the specific sample for the final analysis, which is the main part of the study.
- Further, not all the variables reported in Table 1, 2a, 2b, and 2c are used in multilevel models. For example, perceived birth size, birth order, Took vitamin A supplements are all missing in model 2-4 in Table 4. This is one of the reasons that the manuscript is lengthy. It looks like almost all variables available in the data set were reported then only a few were used in the final analysis. Though it provides interesting and useful information about the status of a variety of factors, it is more likely to confuse readers, interfere the main goals and clarity of the study.
- In Table 2b, what is reported for “Mother’s body mass index(kg/m2)” is body weight status, not BMI.
- Table 3 does not seem to be necessary. It is explained enough in text. Please consider removing it.
- “This group of children, a sub-sample of under-five years forms the unit of analysis for this study.” In line 132-133 can be removed. It is already specified in 2.3.
- “(the default technique in Stata)” on line 209 can be removed.
- For BMI classification, please consider using the term “healthy” instead of “normal” (https://www.cdc.gov/obesity/adult/defining.html).
- Please double-check variable labels across tables. They are not consistent.
- Possible typos: “adjest” on line 195; “[P01]” on line 199; “[PO2]” on line 204.
Author Response
We deeply appreciate you for yet a more extensive and constructive review of the paper. In fact, responding the queries raised have added more facts to enhance the quality of the paper

Round 3
Reviewer 5 Report
Thank you for making changes based on the comments. I believe the manuscript has been significantly improved now.